# Microbiota-directed fibre activates both targeted and secondary metabolic shifts in the distal gut

Leszek Michalak[1], John Christian Gaby[1✉], Leidy Lagos[2], Sabina Leanti La Rosa[1], Torgeir R. Hvidsten[1], Catherine Tétard-Jones[3], William G. T. Willats[3], Nicolas Terrapon[4,5], Vincent Lombard[4,5], Bernard Henrissat[4,5,6], Johannes Dröge[7], Magnus Øverlie Arntzen[1], Live Heldal Hagen[1], Margareth Øverland[2], Phillip B. Pope[1,2,8✉] & Bjørge Westereng[1,8✉]

Beneficial modulation of the gut microbiome has high-impact implications not only in humans, but also in livestock that sustain our current societal needs. In this context, we have tailored an acetylated galactoglucomannan (AcGGM) fibre to match unique enzymatic capabilities of *Roseburia* and *Faecalibacterium* species, both renowned butyrate-producing gut commensals. Here, we test the accuracy of AcGGM within the complex endogenous gut microbiome of pigs, wherein we resolve 355 metagenome-assembled genomes together with quantitative metaproteomes. In AcGGM-fed pigs, both target populations differentially express AcGGM-specific polysaccharide utilization loci, including novel, mannan-specific esterases that are critical to its deconstruction. However, AcGGM-inclusion also manifests a "butterfly effect", whereby numerous metabolic changes and interdependent cross-feeding pathways occur in neighboring non-mannanolytic populations that produce short-chain fatty acids. Our findings show how intricate structural features and acetylation patterns of dietary fibre can be customized to specific bacterial populations, with potential to create greater modulatory effects at large.

[1] Faculty of Chemistry, Biotechnology and Food Science, Norwegian University of Life Sciences, 1432 Ås, Norway. [2] Faculty of Biosciences, Norwegian University of Life Sciences, 1432 Ås, Norway. [3] School of Natural and Environmental Sciences, Newcastle University, Newcastle upon Tyne, UK. [4] Centre National de la Recherche Scientifique, Aix-Marseille Université, UMR7257 Marseille, France. [5] Institut National de la Recherche Agronomique, USC1048 Architecture et Fonction des Macromolécules Biologiques, Marseille, France. [6] Department of Biological Sciences, King Abdulaziz University, Jeddah, Saudi Arabia. [7] Department for Mathematical Sciences, Chalmers University of Technology, Gothenburg, Sweden. [8] These authors contributed equally: Phillip B. Pope, Bjørge Westereng. ✉email: john.christian.gaby@nmbu.no; phil.pope@nmbu.no; bjorge.westereng@nmbu.no

Microbiota-directed foods (MDFs) have emerged as a strategy to modulate the gut microbiome, as the diet has distinct and rapid effects on microbiome composition and function[1,2]. MDFs by definition are not broadly metabolized, but rather elicit a targeted metabolic response in specific indigenous microbiota that confers benefits to their host. This in itself presents a challenge; as many newly identified MDF target organisms, such as beneficial butyrate-producing (i.e., butyrogenic) *Roseburia* and *Faecalibacterium* spp.[3,4], have broad metabolic capabilities that are shared with the vast majority of fiber-fermenting microbiota in the gut ecosystem. Nevertheless, recent studies have revealed intimate connections between the enzymatic and mechanistic features of microorganisms and the glycan structures of the fibers they consume[5,6], which creates new conceptual MDF targets. This is exemplified by discoveries of sophisticated polysaccharide-degrading apparatuses that enable certain microbiota to consume fiber in a 'selfish' manner, whereby complex glycan structures (such as β-mannans) are cleaved into large oligosaccharides at the cell surface, which is subsequently transported into the cell and depolymerized into monomeric sugars[5,7,8]. Such a mechanism restricts the release of sugars into the ecosystem for neighboring scavenging populations, thus giving a selective metabolic advantage to the selfish degrader in the presence of these highly complex glycans.

Beta-mannans are present in human and livestock diets, and depending on their plant origins, can be decorated with varying amounts of acetylation that protect the fiber from enzymatic degradation[9]. We recently demonstrated that the human gut commensal *Roseburia intestinalis* encodes a mannan-specific polysaccharide utilization locus (PUL), and 'selfishly' converts highly complex mannan substrates to butyrate[5]. Within this mannan PUL, a carbohydrate esterase (CE) family 2 (*Ri*CE2) removes 3-*O*-, and 6-*O*-acetylations on mannan, whereas a novel CE family 17 (*Ri*CE17) removes the axially oriented 2-*O*-acetylations[9], which are distinctive features found in limited mannan moieties and inaccessible to most of the characterized bacterial esterases present in the gut microbiome. Closer genome examinations have revealed that putative CE2/CE17-containing mannan PULs are in fact prominent within many butyrate-producers including *Roseburia* spp., *Faecalibacterium prausnitzii*, *Ruminococcus gnavus*, *Coprococcus eutactus* and *Butyrivibrio fibrisolvens*[5,10]. It is well known that the metabolic attributes of these populations are highly desirable in the gastrointestinal tract, and that their depletion is implicated in colorectal cancer, Crohn's disease, inflammatory bowel syndrome, ulcerative colitis, forms of dermatitis, and several other diseases[11,12]. These collective findings thus raised the question: could a custom MDF fiber that was tailored to match these specialized enzymatic capabilities selectively engage butyrate-producers in a complex microbiome ecosystem?

2-*O*-acetylated mannans are found in a limited number of characterized western dietary fiber sources (i.e., tomatoes[13] and coffee[5]), however, 2-*O*-acetylations are present in acetylated galactoglucomannan (AcGGM), which is the main hemicellulose in the secondary cell wall of Norway spruce (*Picea abies*)[14]. We have utilized a controlled steam explosion (SE), followed by ultrafiltration (UF) fractionation to extract complex AcGGM from spruce wood. Processing conditions were selected to tailor the fiber with a high degree of galactose branching's and 2-*O*-, 3-*O*- and 6-*O*-acetylations[15], which is amenable to inclusion as an MDF in animal feed production. Previously, the MDF concept has matched polysaccharides with *Bacteroides*-encoded PULs to demonstrate the creation of exclusive metabolic niches[1,16]. In particular, Shepherd et al.[16] engrafted exogenous strains in mice via their rare PUL-encoded enzymatic capabilities, whereas Patnode et al.[1] illustrated that bioactive carbohydrates were found to target

particular *Bacteroides* species in a defined consortium of 20 human gut microbial species in gnotobiotic mice. However, what is less understood from MDF studies to date, is (1) can the MDF concept be applied to target indigenous populations within a complex endogenous microbiome, and (2) what are the broader secondary community effects if a targeted population is stimulated, i.e., are new niches created and/or existing ones closed?

Here, we test whether our AcGGM fiber can specifically target beneficial Firmicutes species *Roseburia* and *Faecalibacterium* within a 'real-world' gut ecosystem that consists of 100–1000's of different species. To evaluate this, we analyze the gut microbiomes of weaned piglets fed diets containing varying AcGGM levels over a 28-day period that extended from their first meal after sow separation until an adapted, fiber-degrading microbiome was established. Using metagenomics, we monitor temporal changes in the microbiome and phylogenetically and functionally resolve the genomes of indigenous microbiota. In parallel, our detailed quantitative metaproteomic and carbohydrate microarray analyses reveal the metabolic and enzymatic responses of the different microbiota to the varying AcGGM exposure. We demonstrate how the activity of specific beneficial microbiota can be directly stimulated while simultaneously deciphering the secondary, trophic effects on other populations and metabolic niches, with both aspects having broader implications for developing strategies to effectively modulate the gut microbiome.

## Results

**Production of highly complex dietary mannan fibers from wood.** Spruce galactoglucomannan consists of a backbone of β-(1,4)-linked mannose and glucose residues, decorated with α-(1,6) linked galactose branching, and a large degree of esterification of the mannose residues by 2-*O*- and 3-*O*- and 6-*O*-acetylations[14] (Fig. 1a). A crucial part of this study was the development of an efficient, large-scale extraction process entailing SE as well as ultra- and nanofiltration, which ultimately provided high quantities at high purity whilst not damaging the complexity of the AcGGM fiber (Fig. 1b, c). A total of 700 kg of dry Norway spruce chips was processed using SE at conditions corresponding to a combined severity factor ($R'_0$) of 1.70. We produced 50 kg of oligo/polysaccharides for feed production (Fig. 1c–e), with a monosaccharide (Man:Glc:Gal) ratio of 4:1:0.6. The degree of polymerization (DP) of the AcGGM fiber ranged from β-manno-oligosaccharides with DP of 2–10 to manno-polysaccharides (DP of ≥11), with both exhibiting degrees of acetylation (DA = 0.35). Crucially, this DA value was higher than previous iterations of the fiber (DA = 0.28)[7], and its acetylation patterns (previously determined[9]) matched the enzymatic capabilities of mannan PULs encoded in human gut *Roseburia* and *Faecalibacterium* spp.[5,10]. We, therefore, predicted that our AcGGM fiber would match representatives of the same populations that are indigenous to porcine gut ecosystems[17,18] (Fig. 1f).

**AcGGM altered the gut microbiome of weaned piglets.** We previously demonstrated that varieties of AcGGM can be metabolized by pure cultures of *Roseburia intestinalis* L1-82[5], in vitro enrichments with human gut butyrate-producers[10] and 'minimicrobiota' within gnotobiotic mice[5]. Here, we wanted to test our AcGGM fiber's accuracy and ability to elicit a specific response in indigenous representatives of our target populations within a highly complex and competitive endogenous microbiome. In total, four separate cohorts of twelve weaned piglets were given a pelleted basal feed semi-ad libitum, which contained either 0% (control), 1, 2 or 4% AcGGM to additionally determine the level necessary to elicit an effect on both the host and its microbiome.

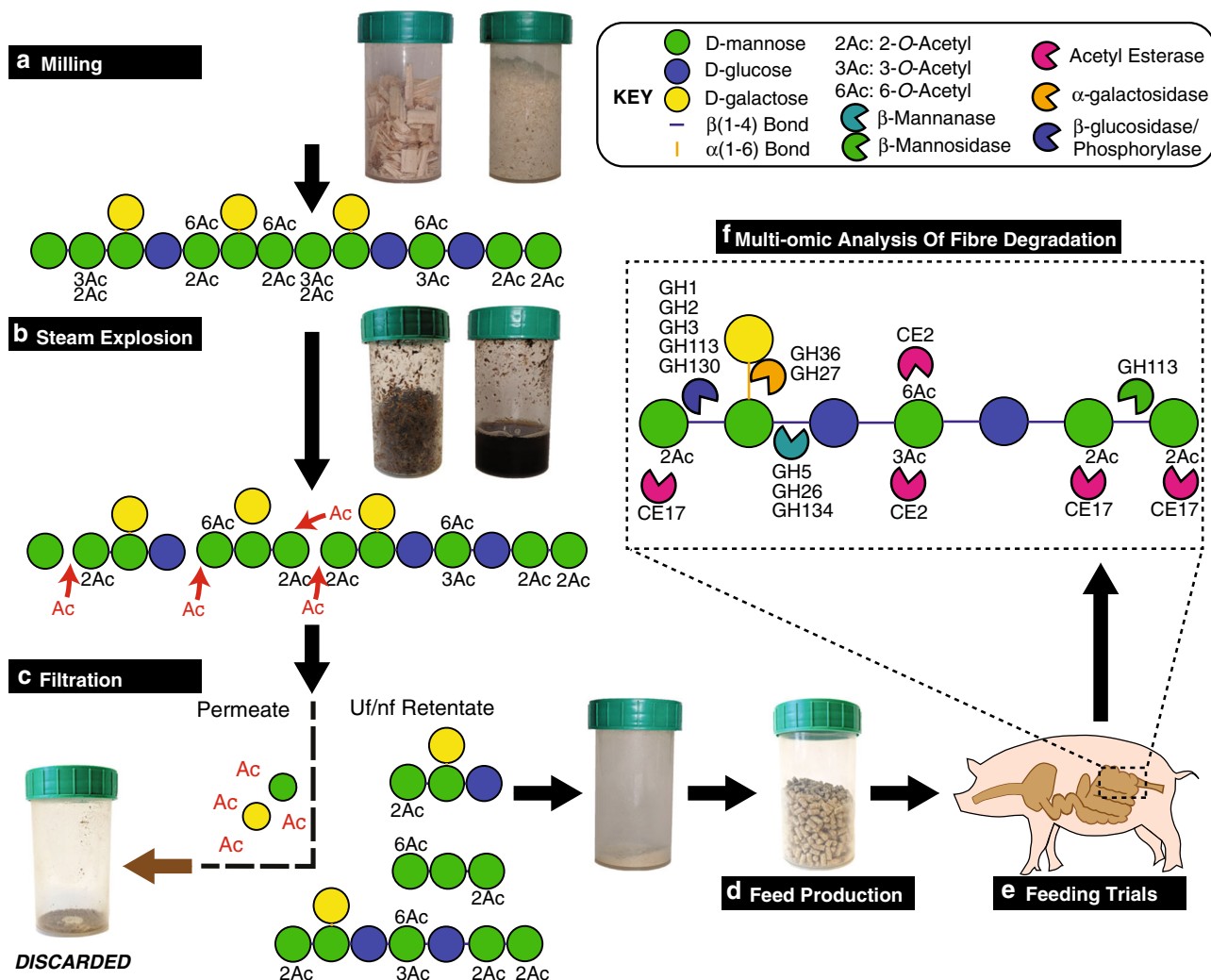

**Fig. 1 Schematic representation and graphic illustration of the production pipeline for Norway spruce AcGGM. a** Wood chips were milled to increase the surface area exposed for hydrothermal extraction and washing the released manno-oligosaccharides. Spruce wood contains long-chained galactoglucomannan (DP200-400), which is highly acetylated (DA ~ 0.65), predominantly with 2-*O* and 3-*O* acetylations[66]. **b** During hydrothermal pretreatment, the release of acetic acid promoted hydrolysis of glycosidic bonds, thereby relinquishing monosaccharides, oligosaccharides, and other breakdown products. **c** Ultrafiltration retained the longer, complex oligosaccharides and discarded the monosaccharides, acetic acid, and other steam explosion byproducts. **d** The purified mannan was incorporated into feed pellets at varying inclusion levels, produced by a conventional feed pelleting process. **e** The growth performance experiment and a feeding trial were conducted in a randomized block design, with four inclusion levels of AcGGM. **f** Multi-omic approaches were used to analyze the porcine gut microbiome in response to AcGGM and determine if indigenous mannan PULs matched the glycan structure of the AcGGM fiber. Glycosidic bonds between the β-(1,4)-mannose and glucose units in the backbone of AcGGM require hydrolysis by glycoside hydrolases (GH) from families GH5 and GH26. GH36 and GH27 α-galactosidases are required to remove the α-(1,6)-galactose decorations. Single mannose and glucose units are removed from the non-reducing end of the oligosaccharides by enzymes from families GH1, GH2, GH3, GH113, and GH130, while mannose in the reducing end can be removed by GH113 family mannosidases. 3-*O*- and 6-*O*-acetylations on mannan are removed by esterases from family CE2. A unique feature of particular beta-mannans is the axially oriented 2-*O*-acetylation on mannose, which is the prevalent form of acetylation present on AcGGM used in this study. 2-*O*-acetylations are removed by esterases homologous to the RiCE17 mannan-specific esterase from *Roseburia intestinalis*, which was recently characterized by our group[5].

We chose a dietary supplementation strategy, which is commonly applied to piglets during the post-weaning period to offset their particularly low feed intake and enhance gut health and growth performance[19]. Cautious measures were made to eliminate potential pen biases (see details in materials and methods), and fecal samples, as well as animal performance metrics, were taken before AcGGM administration (when piglets were assigned to pens), and subsequently at days 7, 14, 21 and 27 during the feeding trial. On day 28, the piglets were sacrificed and host gut tissue and digesta samples taken from all intestinal regions (duodenum, jejunum, ileum, cecum, and colon) for down-stream analysis (Supplementary Table 1 and Supplementary Data 1).

Measurements of major short-chain fatty acids (SCFAs) in the cecum and colon showed a trend of incremental increases of absolute and relative butyrate levels as AcGGM levels were increased (Fig. 2). However, despite SCFA data suggesting that AcGGM inclusion promotes butyrogenic fermentation, it also showed that there was no statistically significant increase in total SCFA levels (Supplementary Data 1). Similarly, the levels of propionic acid were not affected by AcGGM inclusion (Supplementary Data 1). While changes in SCFA and microbiome composition (Figs. 2 and 3) resulted from AcGGM inclusion, we observed no adverse effects on the host's physiology, with the average weight, feed conversion ratio, blood cell composition, T

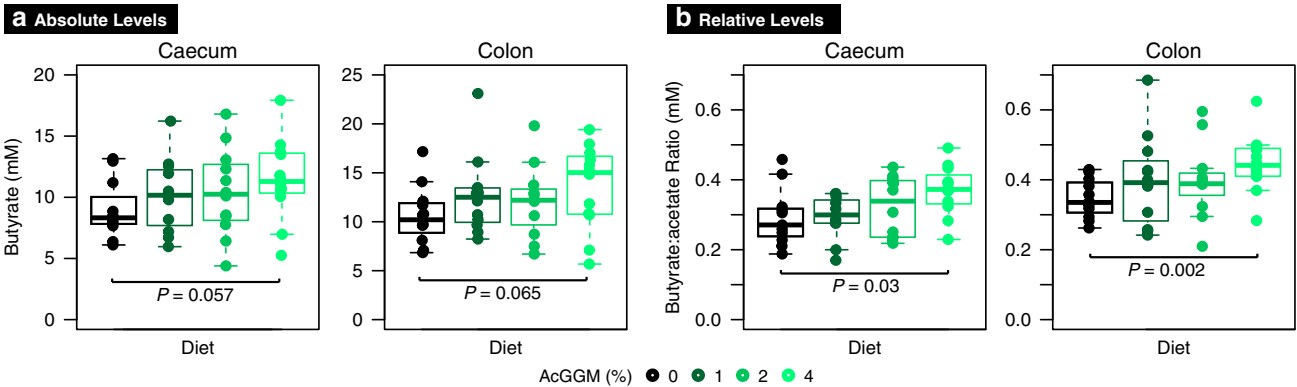

**Fig. 2 Butyrate detected in the cecum and colon digesta of pigs fed the four different AcGGM diets with varying inclusion levels (0–4%). a** display absolute levels and **b** relative levels of butyrate. Raw SCFA measurements are presented in Supplementary Data 1. Bars represent median and boxes interquartile range determined from 12 animals analyzed per dietary group. Data were analyzed using a two-tailed $t$-test, with $p$-values indicated between pigs fed either the control or 4% AcGGM diets. The boxes span the 25th–75th percentiles with the central bars being the medians. Whiskers extend maximum up to 1.5× the interquartile range (IQR) or, when all values are within 1.5× IQR, then the whisker extends to the most extreme data point.

cell population, and colon morphology not differing between the control and AcGGM treatments (Supplementary Fig. 1 and Supplementary Data 1).

Monitoring of temporal microbiome changes using 16S rRNA gene analysis over the month-long trial indicated that the inclusion of AcGGM into the piglets feed caused a pronounced shift in the microbiome structural composition from the 21st day of the trial onwards (Supplementary Figs. 2b and 3). Spatial changes were also examined at the final sampling day and showed typical patterns, whereby the structure of the gut microbiome varied by gut region (Supplementary Figs. 2a, 3, and 4). As expected, the AcGGM-effect was more pronounced in the fiber-fermenting distal regions (cecum, colon) of the gut, where relative abundance of hundreds of phylotypes was observed to change (adjusted $p < 0.05$) in response to varying inclusion levels (Fig. 3a and Supplementary Data 2). To determine the effect AcGGM had on microbiome function, we also closely examined the 355 metagenome-assembled genomes (MAGs, Supplementary Fig. 5a) that were reconstructed from metagenomic data generated from the colon samples of each pig fed the control and 4% AcGGM diets (Supplementary Table 2). The taxonomic affiliation of the MAGs was calculated via GTDB-Tk (Supplementary Data 3), while phylogeny was inferred from a concatenated ribosomal protein tree (Newick format available in Supplementary Data 4) that was constructed using MAGs from this study and 293 closely related reference genomes.

Our target butyrogenic populations produced mixed results, whereby the 16S rRNA gene relative abundance of *Faecalibacterium* affiliated phylotypes increased in response to increasing levels of AcGGM (Fig. 3c and Supplementary Fig. 3c), whereas *Roseburia*-affiliated phylotypes seemingly decreased (Fig. 3b and Supplementary Fig. 3b). However, a detailed analysis of *Roseburia*-affiliated MAGs (GTDB-Tk assigned as *Agathobacter*, see Fig. 4) showed that specific phylotypes that encoded AcGGM-specific PULs were indeed stimulated by the AcGGM fiber (Fig. 3b). Reputable fiber-fermenting populations affiliated to *Prevotella*[20] also showed varying responses (Fig. 3g), with 16S rRNA gene relative abundance of individual phylotypes increasing from 4 to 12% between the control and 4% AcGGM inclusion in both colon and cecum (Supplementary Fig. 3a). Interestingly, relative abundance estimates of both 16S rRNA gene OTUs and MAGs indicated that phylotypes affiliated to non-fiber-degrading taxa, such as *Catenibacterium*[21], *Dialister*[22], and *Megasphaera*, demonstrated some of the highest dose-dependent increases in relative abundance in response to AcGGM (Fig. 3d–f), indicating

that other underlying factors are likely dictating microbiome structure, besides fiber degradation.

**Targeted mannan PULs were detected in the colon of AcGGM-fed pigs.** Because our primary goal was to elucidate whether our target, butyrogenic populations were activated in response to AcGGM, we annotated MAGs affiliated to butyrate-producers and conducted the metaproteomic analysis with label-free quantification (LFQ) on randomly selected colon samples from four control and four 4% AcGGM-fed pigs (Fig. 4a, b), and mapped 8515 detected protein groups back to our MAGs to identify functionally active populations (Fig. 4c) (Supplementary Data 5 and 6). Community-wide analysis of the MAG genetic content (Supplementary Fig. 5b) from each sample and clustering analysis of their detected proteins (Fig. 4a, b) further supported our 16S rRNA gene analysis, reiterating that the microbiomes from piglets fed the control and 4% AcGGM diets were distinct.

Our MAG-centric multi-omic approach gave clear indications as to what effect the AcGGM fiber had on putative butyrogenic *Roseburia* and *Faecalibacterium* populations in the distal gut of pigs. Fifteen MAGs clustered with representative *Roseburia* spp. genomes (Figs. 3b and 4), which reflected the multiple *Roseburia*-affiliated phylotypes that were predicted with our 16S rRNA gene analysis (Supplementary Fig. 3b and Supplementary Data 2). In general, the relative abundance of *Roseburia*-affiliated MAGs (Fig. 3b) and detected proteins (Fig. 4c) were observed at either static or lower levels in AcGGM-fed pigs, reiterating our initial 16S rRNA gene observations that AcGGM negatively affected *Roseburia* populations (Fig. 3b). However, one specific *Roseburia*-affiliated population (MAG041) was detected at significantly higher abundance levels ($p = 0.0016$) (Fig. 3b), and its detected proteins were enriched (adj. $p = 0.0034$) in the 4% AcGGM pig samples compared to the control (Fig. 4b, c and Supplementary Data 7).

Closer examination of MAG041 revealed a putative CE2/CE17-containing mannan-degrading PUL that was absent in the other *Roseburia*-affiliated MAGs and was differentially expressed in the AcGGM diet (Fig. 5). Importantly, the MAG041 mannan PUL exhibited gene synteny to the *R. intestinalis* strain L1-82 PUL whose biochemical properties we recently characterized in detail[5] (Fig. 5). The predicted multi-modular mannanase (CBM27-GH26-CBM23) in the MAG041 mannan PUL is homologous to the GH26 in *R. intestinalis* L1-82 (48% identity over 87% of the sequence), and can be presumed to fulfill the same function—'selfishly' breaking down AcGGM fibers at the cell surface prior to intracellular transport.

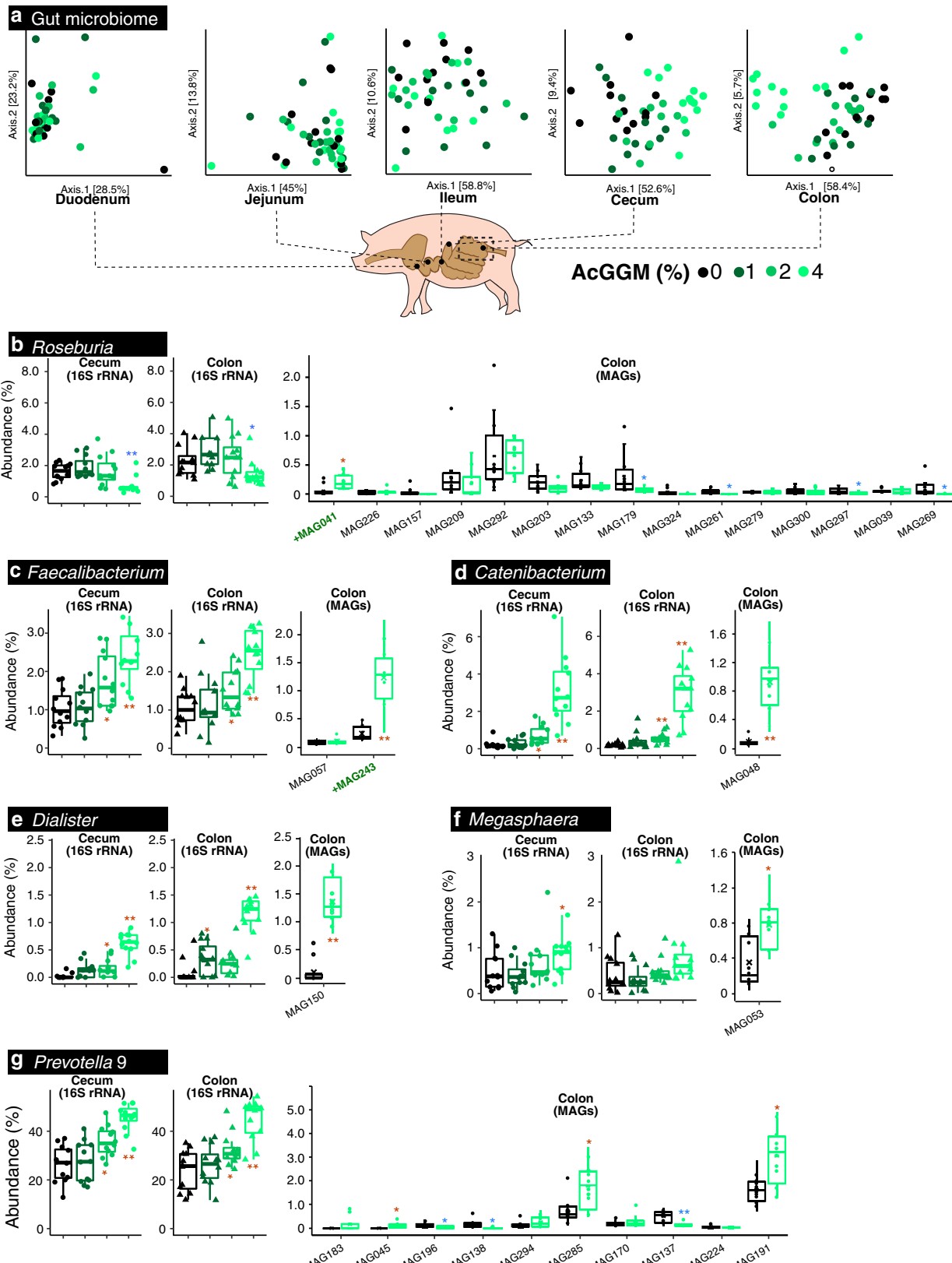

Besides the detection of GH26 and esterases in AcGGM-fed pigs, other mannan-specific enzymes also responded to the dietary shift within the MAG041 mannan PUL, including a phosphoglucomutase, a multiple-sugar binding protein, a GH130.1 4-O-β-D-mannosyl-D-glucose phosphorylase and a GH130.2 β-1,4-manno-oligosaccharide phosphorylase (Fig. 5).

In contrast to *Roseburia*-affiliated MAGs, only one MAG clustered with *F. prausnitzii* (MAG243, Fig. 4 and Supplementary Data 3), implying that the multiple phylotypes that were predicted with our 16S rRNA gene data (Supplementary Fig. 3c and Supplementary Data 2) encode high genome similarity and coverage and thus likely co-assembled into a representative

**Fig. 3 Effect of AcGGM-containing diets on the pig gut microbiome.** 16S rRNA gene amplicon and shotgun metagenomic analysis was used to monitor the effect the AcGGM fiber had on the gut microbiome structure of weaned piglets, determined from 12 animals analyzed per dietary group. Both 16S rRNA and MAG relative abundance data were analyzed using a two-tailed *t*-test, with significant increases (red) and decreases (blue) indicated between pigs fed either the control or AcGGM diets (* denotes *p* < 0.05, ** denotes *p* < 0.001). The boxes span the 25th–75th percentiles with the central bars being the medians. Whiskers extend maximum up to 1.5× the interquartile range (IQR) or, when all values are within 1.5× IQR, then the whisker extends to the most extreme data point. **a** Ordination plots of Bray–Curtis distances between microbial communities from pigs feed either the control or AcGGM diets (at varying inclusion levels: 1, 2 or 4%, 12 animals per dietary group). Samples were collected at day 28 of the feeding trials from various sections of the small and large intestine. The AcGGM-effect was more pronounced in the fiber-fermenting distal regions (cecum, colon) of the gut. **b** Relative 16S rRNA gene abundance of *Roseburia*-affiliated phylotypes (agglomerated at the genus level) in either cecum or colon samples collected from pigs fed AcGGM diets with varying inclusion levels (cecum: 0% vs 4% *p* = 0.0004, colon: 0% vs 4% *p* = 0.034). MAG relative abundances were determined from colon samples only, collected from pigs fed either the control or 4% AcGGM diets (0% vs 4% *p*-value: MAG041 = 0.002, MAG179 = 0.028, MAG261 = 0.001, MAG297 = 0.014, MAG269 = 0.035). Despite being suspected as an active mannan degrader, 16S rRNA gene abundance of *Roseburia*-affiliated phylotypes decreased with increasing % AcGGM. However, MAG relative abundance analysis of *Roseburia* populations showed that specific phylotypes that encoded AcGGM-specific PULs (indicated by green text) were indeed stimulated by the AcGGM fiber. **c** The relative 16S rRNA gene abundance of the genus *Faecalibacterium* was enhanced by the inclusion of AcGGM (cecum: 0% vs 2% *p* = 0.008, 0% vs 4% *p* = 2.11E−05, colon: 0% vs 2% *p* = 0.046, 0% vs 4% *p* = 4.64E−06), as were affiliated MAGs in the colon (0% vs 4% *p*-value: MAG243 = 5.84E−06). The same trends were observed for phylotypes affiliated to *Catenibacterium* (**d** cecum: 0% vs 2% *p* = 0.006, 0% vs 4% *p* = 1.12E−05, colon: 0% vs 2% *p* = 0.0007, 0% vs 4% *p* = 1.08E−06, MAG048 = 1.52E−06), Dialister (**e** cecum: 0% vs 2% *p* = 0.013, 0% vs 4% *p* = 3.29E−09, colon: 0% vs 1% *p* = 0.032, 0% vs 4% *p* = 1.82E−07, MAG150 = 2.07E−09), Megasphaera (**f** cecum: 0% vs 4% *p* = 0.031, colon: 0% vs 4% MAG053 = 0.0011) and Prevotella group 9 (**g** cecum: 0% vs 2% *p* = 0.003, 0% vs 4% *p* = 4.71E−07, colon: 0 vs 2% *p* = 0.010, 0% vs 4% *p* = 3.96E−06, MAG045 = 0.008, MAG196 = 0.010, MAG138 = 0.0018, MAG285 = 0.0024, MAG137 = 6.99E−05, MAG191 = 0.0014). Due to space constraints, only a partial representation of Prevotella-affiliated MAG abundances is illustrated in **g**.

population-level MAG. The relative abundance of MAG243 in the distal gut of pigs fed 4% AcGGM increased ~5-fold to ~1.24% (*p* = 5.84E−06) (Fig. 3c), while metaproteomic hierarchical clustering analysis showed that MAG243 proteins were enriched in the same samples (Fig. 4b, c and Supplementary Data 7). Similar to MAG041, MAG243 was found to encode a CE2/CE17-containing mannan PUL, which was broadly detectable in the presence of AcGGM but absent in the control samples (Fig. 5 and Supplementary Data 6). While the MAG243 mannan PUL contained two GH130 manno-oligophosphorylases, a mannose 6-phosphate isomerase, phosphoglucomutase, and two carbohydrate esterase (CE17 and CE2), it lacked a GH26 mannanase representative, which suggests that *F. prausnitzii* is likely preferentially targeting the shorter acetylated manno-oligosaccharides that form part of the AcGGM structure (Fig. 1). In addition to the mannan PULs of MAG041 and MAG243 being activated in AcGGM-fed pigs, their butyrogenic pathways were also detected at high levels, based on label-free quantification (LFQ) scores of detected proteins (Figs. 6, 7 and Supplementary Data 6), suggesting that both populations can convert mannan to butyrate (Supplementary Data 6).

**Specific removal of acetylations is key to access AcGGM.** A crucial step in the utilization of mannans as an energy source is the deacetylation of 2-*O*-, 3-*O*- and 6-*O*-mannose residues, which allows the subsequent breakdown of the sugar-containing backbone of the fiber. In *R. intestinalis* L1-82, AcGGM deacetylation occurs via the synergistic actions of two carbohydrate esterases (*Ri*CE2 and *Ri*CE17) that exert complementary specificities[9]. MAG041 and MAG243 both encoded CE2 homologs within their mannan PULs, sharing 63 and 31% identity (respectively) to *Ri*CE2, which has demonstrated activity on 3-*O*-, (4-*O*-) and 6-*O*-acetylations, and is mannan specific[9]. For CE17, MAG041 and MAG243 homologs shared 65 and 46% identity (respectively) with *Ri*CE17, including the active site residues and the aromatic stacking tryptophan (Trp326), which in *Ri*CE17 are associated with 2-*O*-acetylation specificity[9]. Broader screens of our MAG data revealed other CE2/CE17-containing PULs within Firmicute-affiliated MAGs from the pig colon microbiome (Supplementary Fig. 6), however, aside from MAG041 and MAG243, they originated from populations whose MAG relative abundance was very low (<0.05%) and metabolic activity was

undetectable via metaproteomics in any of the control or AcGGM diets (Fig. 4). Finally, the differential proteomic detection of MAG041 and MAG243 CEs in pigs fed AcGGM diets (Fig. 5), strengthened our hypothesis that both these populations can accommodate the unique features of the AcGGM fiber and are actively engaging in its utilization in vivo.

**AcGGM also causes an effect in non-target populations.** MDFs studies to date have eloquently highlighted that metabolic symmetry between individual fibers and microbiota can be used to stimulate specific populations[1,16]. However, deeper microbiome effects that result from a target species being enriched via MDFs are poorly understood. Although the specificity of the AcGGM fiber matched selected mechanistic features of our target populations, our data showed that AcGGM dietary intervention reverberated further down the microbial trophic networks that support the conversion of dietary fiber into keystone SCFAs that are of nutritional value to the host animal. In particular, we observed varying effects upon different non-AcGGM-degrading butyrogenic populations belonging to clostridial cluster XIVa (Lachnospiraceae: *Roseburia* spp., *Eubacterium rectale*, *Butyrivibrio* and *Pseudobutyrivibrio*) and cluster IV (Ruminococcaceae: *Faecalibacterium prausnitzi*-related bacteria)[23] (Fig. 6 and Supplementary Data 6). Specific populations that were closely related to AcGGM-degrading MAG041 were found to be either unaffected (MAG292) or metabolically suppressed (MAG133) in the presence of 4% AcGGM. Indeed, detected proteins from MAG133 were enriched in pigs fed the control diet (adj. *p* = 2.8E−18, Supplementary Data 7), and enzymes associated with butyrate production were largely undetected in 4% AcGGM-fed pigs (Fig. 6 and Supplementary Data 6). In most cases, non-AcGGM degrading, butyrogenic populations were seemingly utilizing sugars found in starch, arabinoxylan, and/or arabinogalactan fibers that were detected in the basal feed components using Micro Array Polymer Profiling (MAPP; Supplementary Data 8). A broader analysis of our omic data identified multiple abundant populations with similar metabolic capabilities, suggesting that suppressed butyrogenic populations were possibly being outcompeted by Prevotella-affiliated populations such as MAG285 (see below). While SCFA measurements indicated an increase in relative butyrate levels in AcGGM-fed pigs (Fig. 2b), the indirect effects of our MDF on this dynamic functional group are likely

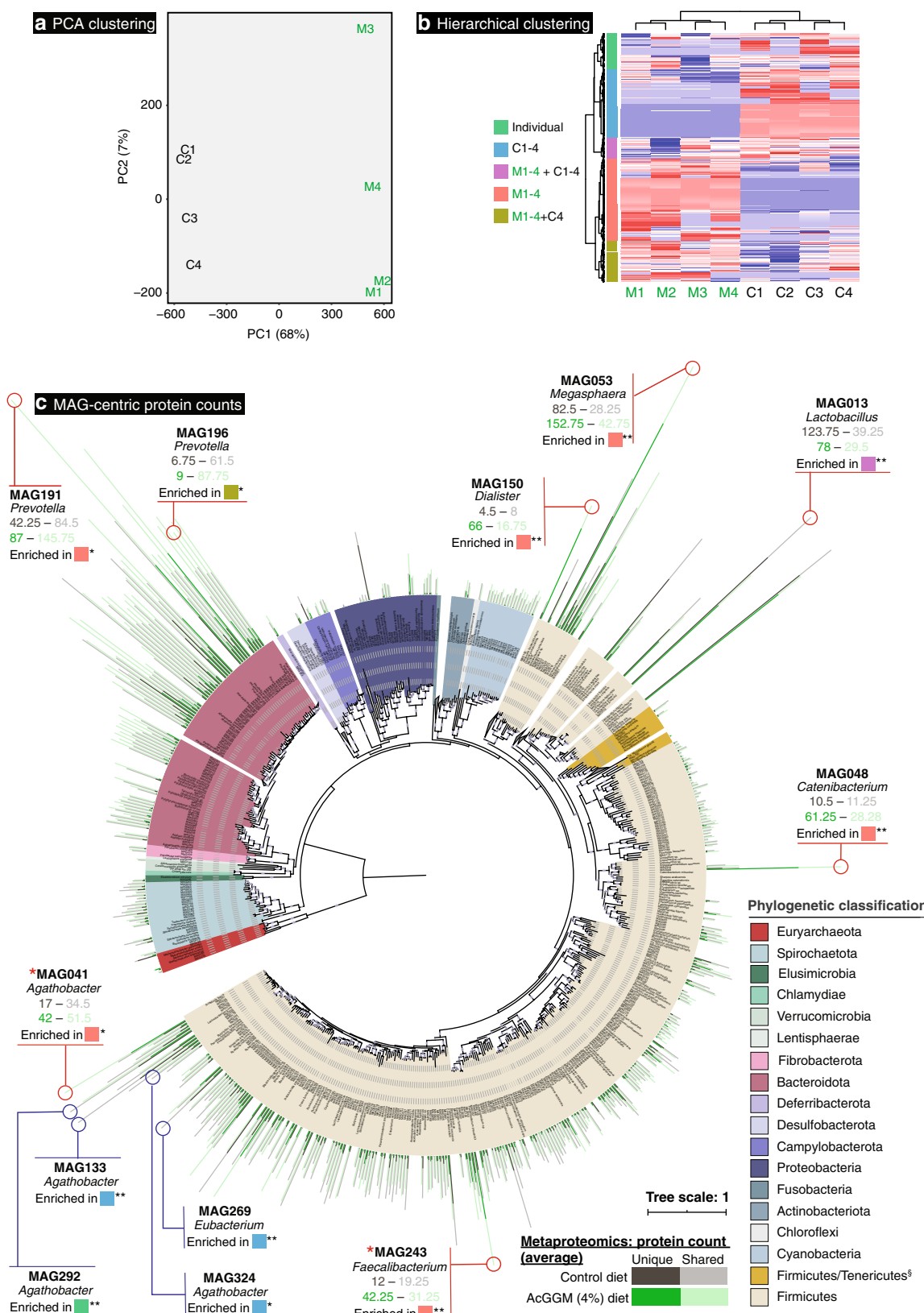

**a** PCA clustering

**b** Hierarchical clustering

**c** MAG-centric protein counts

**MAG191**
*Prevotella*
42.25 – 84.5
87 – 145.75
Enriched in *

**MAG196**
*Prevotella*
6.75 – 61.5
9 – 87.75
Enriched in *

**MAG053**
*Megasphaera*
82.5 – 28.25
152.75 – 42.75
Enriched in **

**MAG013**
*Lactobacillus*
123.75 – 39.25
78 – 29.5
Enriched in **

**MAG150**
*Dialister*
4.5 – 8
66 – 16.75
Enriched in **

**MAG048**
*Catenibacterium*
10.5 – 11.25
61.25 – 28.28
Enriched in **

***MAG041**
*Agathobacter*
17 – 34.5
42 – 51.5
Enriched in *

**MAG133**
*Agathobacter*
Enriched in **

**MAG269**
*Eubacterium*
Enriched in **

**MAG324**
*Agathobacter*
Enriched in *

**MAG292**
*Agathobacter*
Enriched in **

***MAG243**
*Faecalibacterium*
12 – 19.25
42.25 – 31.25
Enriched in **

**Phylogenetic classification**

- Euryarchaeota
- Spirochaetota
- Elusimicrobia
- Chlamydiae
- Verrucomicrobia
- Lentisphaerae
- Fibrobacterota
- Bacteroidota
- Deferribacterota
- Desulfobacterota
- Campylobacterota
- Proteobacteria
- Fusobacteria
- Actinobacteriota
- Chloroflexi
- Cyanobacteria
- Firmicutes/Tenericutes§
- Firmicutes

Tree scale: 1

**Metaproteomics: protein count (average)**

| | Unique | Shared |
|---|---|---|
| Control diet | | |
| AcGGM (4%) diet | | |

dampening an even greater influence of AcGGM-stimulated *Roseburia* and *Faecalibacterium* populations upon the broader microbiome and host animal (Fig. 6).

Besides butyrate-producers, MAG-centric metaproteomic enrichment analysis showed that fiber-degrading *Prevotella*-affiliated populations experienced contrasting effects as a result of AcGGM inclusion. For example, MAG191 was found to account for the highest levels of detectable proteins in our datasets (Fig. 4c and Supplementary Data 6), which were enriched in clusters differently detected in the presence of 4% AcGGM

**Fig. 4 Genome-centric metaproteomic analysis of colon samples collected from pigs fed either the control of 4% AcGGM diet. a** Principle Component Analysis of metaproteomes generated from randomly selected colon samples collected from pigs fed either the control (C1-4) or 4% AcGGM (M1-4) diet. A clear separation was observed between the two diets, highlighting that the detected proteins in the colon microbiomes from piglets fed the control, and 4% AcGGM diets were distinct. **b** Hierarchal clustering and heatmap of detected protein group abundance profiles. Rows are scaled so that red represents the highest abundance for that protein group and blue the lowest. Five different clusters were observed, with protein groups differentially detected in AcGGM-fed pigs (M1-4: red), control pigs (C1-4: blue), all pigs (M1-4 + C1-4: purple), AcGGM-fed pigs plus one control (M1-4 + C4: brown) and only in individual pigs (Individual: green). **c** Phylogeny and metaproteomic detection of 355 MAGs sampled from the colon of weaned piglets. This maximum likelihood tree is based on an alignment of 22 concatenated ribosomal proteins from the 355 MAGs reconstructed in this study from 24 colon metagenomes (12 control pigs, 12 fed 4% AcGGM diet), in addition to 239 reference genomes closely related to the MAGs. Branches are shaded with color to highlight phylum-level affiliations (see legend). Colored bars on the outside of the tree depict the average number of unique and shared protein groups detected for each MAG in four randomly selected samples that were analyzed from pigs fed either the control- (gray) or 4% AcGGM diet (green). Total number of proteins for each MAG in each pig is detailed in Supplementary Data 5. Purple circles on the inside of the tree represent nodes with bootstrap support ≥70%, relative to size. MAG041 and MAG243 were found to encode CE2/CE17-containing mannan PULs (Fig. 5) are indicated by *. All MAGs depicted in Fig. 6 (blue circles) and Fig. 7 (red circles) are listed, with average detected protein counts in both diets, which metaproteomic expression cluster they are enriched in (part **b**, adjusted *p*-values are indicated: * denotes *p* < 0.05, ** denotes *p* < 0.001, exact values listed in Supplementary Data 7), and their MAG taxonomic affiliation determined via GTDB-Tk. MAG enrichment analysis was performed using the hypergeometric distribution function phyper in R with the false discovery rate controlled at 5% using the function *p*.adjust with method = 'BH'. Recently reclassified *Roseburia* species are denoted as Agathobacter. The full tree in Newick format is provided in Supplementary Data 4.

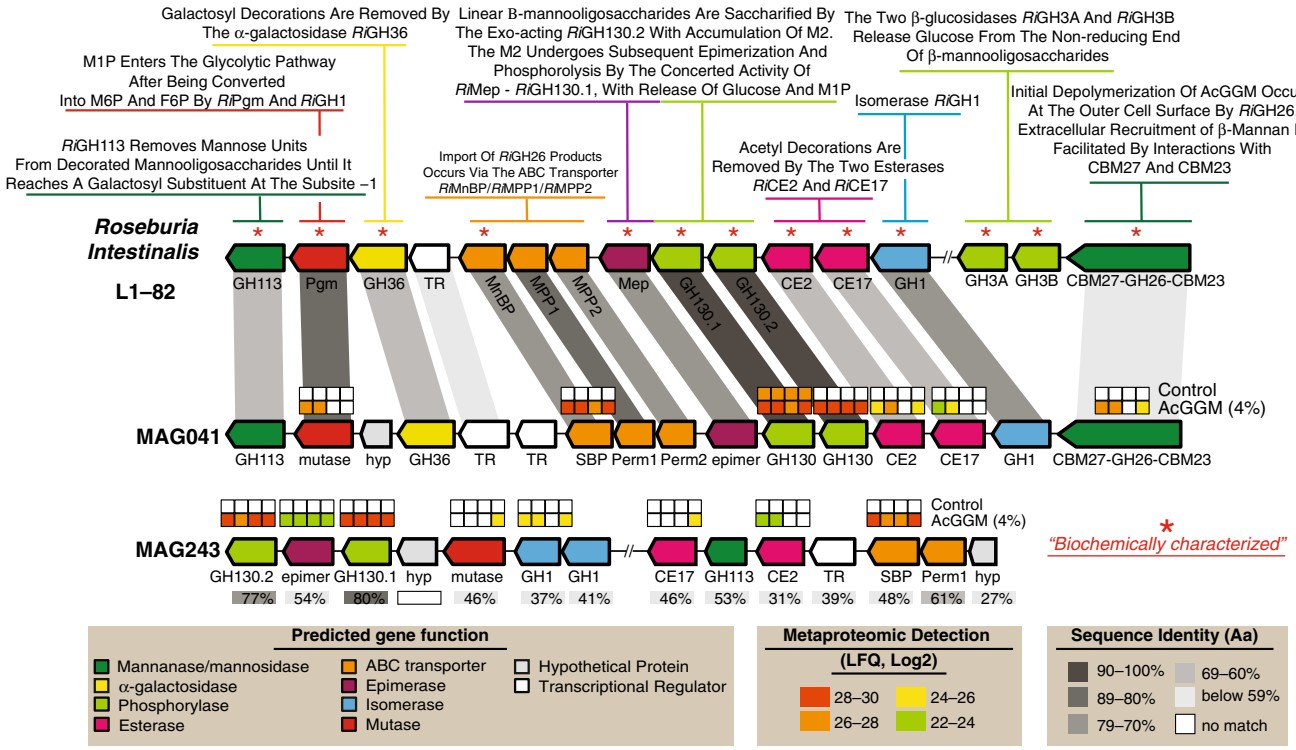

**Fig. 5 Metaproteomic detection of CE2/CE17-containing mannan PULs encoded in *Roseburia*- (MAG041) and *Faecalibacterium*- (MAG243) affiliated MAGs in pigs fed with either the control or 4% AcGGM diet.** Predicted gene organization and annotated gene function is color-coded and largely derived from the previous biochemical and structural characterization of the mannan degradation cluster (characterized genes indicated with *) in *R. intestinalis* L1-82[5]. Gene synteny and identity % between mannan PULs found in *R. intestinalis* L1-82, MAG041 and MAG243 are indicated in gray boxes. Heat maps above detected enzymes show the LFQ detection levels for the four replicates sampled in control and 4% AcGGM-fed pigs. LFQ values of proteins from both clusters are in Supplementary Data 6. The predicted multi-modular mannanase (CBM27-GH26-CBM23) from MAG041 was the only extracellular protein in the locus, and the only extracellular mannanase expressed in response to AcGGM inclusion.

(adj. *p* = 0.0023, Supplementary Data 6). Pathway annotation of AcGGM-enriched *Prevotella* populations (such as MAG191, MAG196, MAG285, see Figs. 4c, 7, and Supplementary Data 6, 7) indicated active metabolism of dietary fibers such arabinoxylans, starch, glucans (e.g., cellobiose), α-galactans, and mannose sugars (detected via MAPP analysis, Supplementary Data 8) as well as acetate, succinate and/or propionate production, which were all detected with higher LFQ scores in AcGGM-fed pigs (Fig. 7 and Supplementary Data 6). However, many CAZymes

and fermentation enzymes from the aforementioned MAGs were also detected across both diets (albeit at varying LFQ values), while other *Prevotella* populations were specifically enriched in pigs fed the control diet (i.e., MAG034: adj. *p* = 2.0E−04, Supplementary Data 7), suggesting that *Prevotella*-driven baseline consumption of basal feed fiber was occurring irrespective of AcGGM inclusion (Fig. 7 and Supplementary Data 6).

Several mannan-targeting PULs were identified in *Prevotella*-affiliated MAGs that were configured in an archetypical

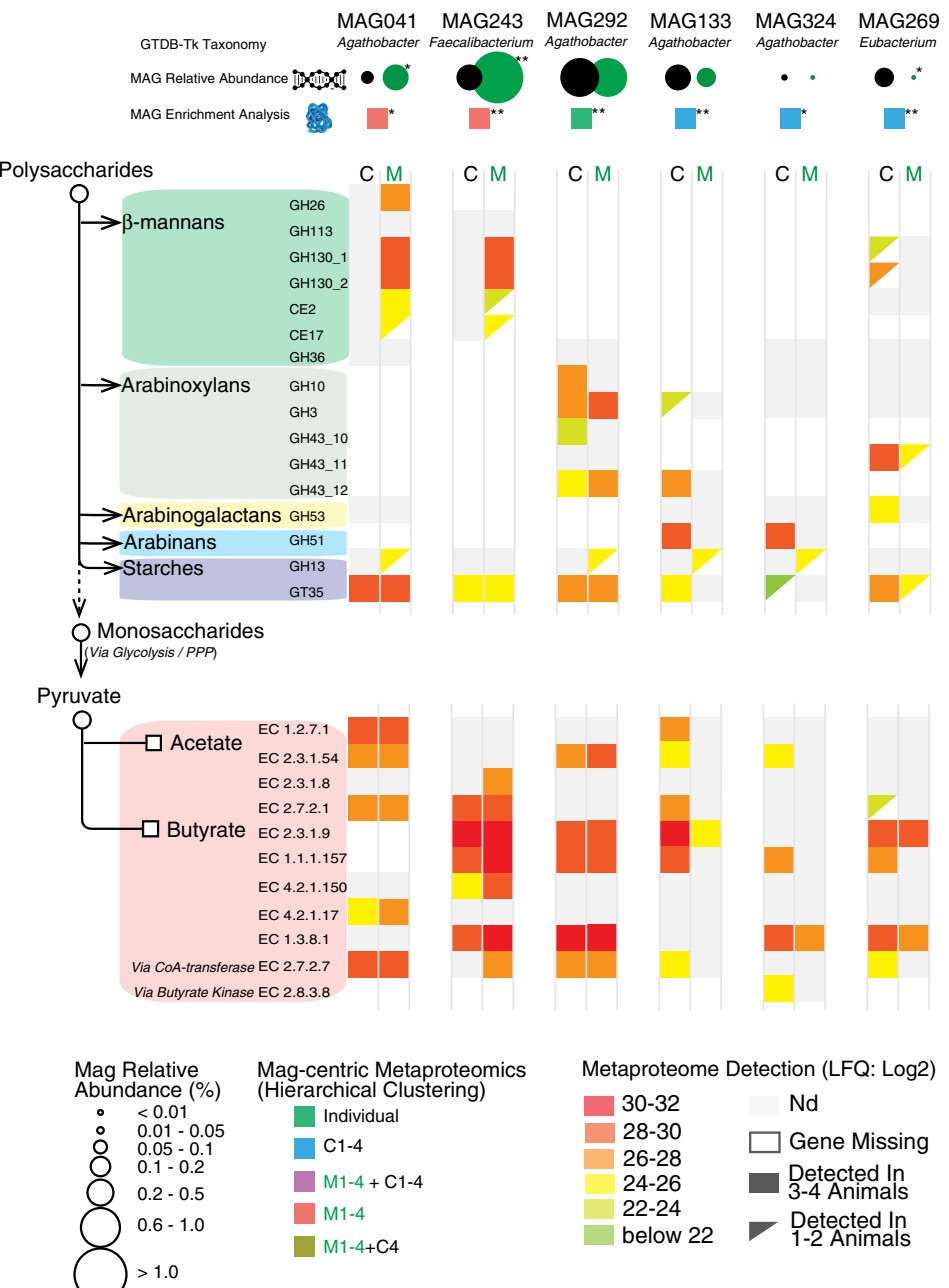

**Fig. 6 The differing metabolisms of butyrate-producers in the colon microbiome of pigs fed AcGGM.** A plethora of MAGs affiliated with putative butyrate-producers was recovered from the pig colon metagenome, which varied in their fiber-degrading capacity and detected activity in either the control (C) or 4% AcGGM (M) diets. Here, we present a representative selection of populations that: encoded specific AcGGM-degrading capabilities (MAG041 and MAG243), were abundant across both diets with no apparent effect (MAG292), were significantly enriched in pigs fed the control diet (MAG133, MAG324, and MAG269), and encoded an alternative pathway for butyrate production (via Butyrate kinase) (MAG324). GTBD-Tk inferred taxonomy (recently reclassified *Roseburia* species are denoted as *Agathobacter*), MAG relative abundance (control: black circles, AcGGM: green circles, *p*-value: MAG041 = 0.002, MAG243 = 5.84E−06, MAG292 = 0.98, MAG133 = 0.051, MAG324 = 0.120, MAG269 = 0.035) and MAG-centric metaproteomic enrichment analysis is indicated (see Fig. 4c, significant differences denoted by adjusted *p*-values: *p < 0.05, **p < 0.001, MAG041 = 3.4E−03, MAG243 = 8.5E−19, MAG292 = 5.2E−06, MAG133 = 2.8E−18, MAG324 = 4.9E−02, MAG269 = 1.9E−05). MAG enrichment analysis was performed using the hypergeometric distribution function phyper in R with the false discovery rate controlled at 5% using the function *p*.adjust with method = 'BH'. CAZymes (GH, CE, and GT) involved in the catabolism of the listed polysaccharides are grouped according to box color and their specific activities are described on www.cazy.org. Enzymes associated with acetate and butyrate metabolic pathways are listed as EC (enzyme commission) numbers. Corresponding functions can be found on https://www.genome.jp/kegg/kegg2.html.

'Bacteroidetes-format', which combines outer-membrane transport and carbohydrate-binding SusC/D-like proteins as well as CAZymes[24] (Supplementary Fig. 6a). In particular, a PUL recovered from MAG196 encoded predicted SusC/D-like proteins, mannanases (GH26, GH5_7), mannosylphosphorylases (GH130), and an esterase, although neither the mannanases nor the esterase was detected in the metaproteomes recovered from the AcGGM-fed pigs (Fig. 7, Supplementary

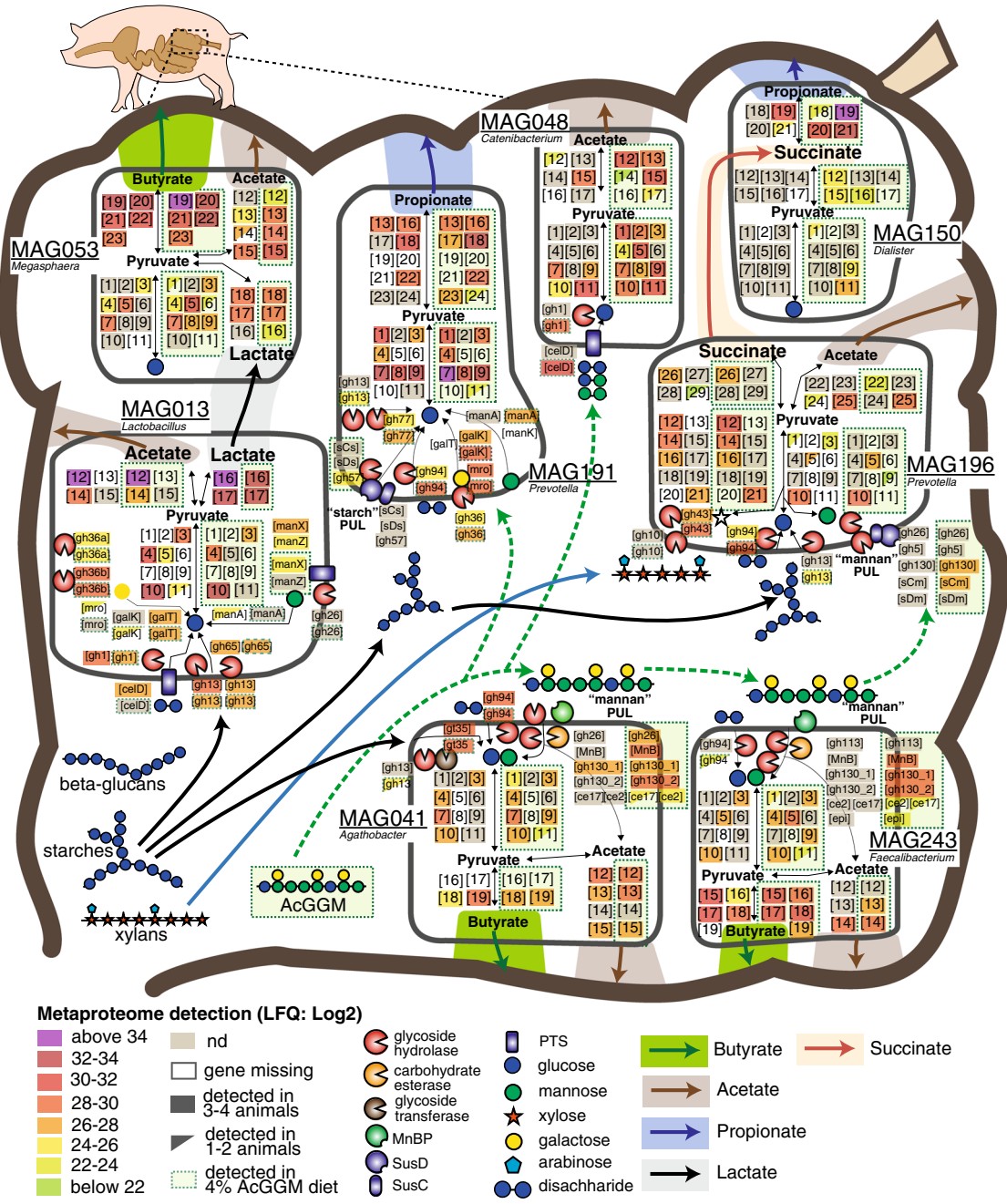

**Fig. 7 Selected metabolic features of the porcine colon microbiome in response to AcGGM dietary intervention, as inferred from genome and proteome comparisons.** The different metabolic pathways (fiber deconstruction, glycolysis, pentose-phosphate pathway, and SCFA production) are displayed for each population MAG. Graphical representation of pathways, enzymes, CAZymes, and cellular features are based on functional annotations that are depicted as numbered or abbreviated gene boxes, which are additionally listed in Supplementary Data 6. Metaproteomic analysis (detected genes and enzyme complexes) is highlighted according to the different LFQ values of the detected proteins sampled at a single time-point (28 days) either from animals fed the control or 4% AcGGM diet (green perforated boxes). The main dietary fibers (starches, xylans, glucans, and mannans), SCFAs (butyrate, acetate, propionate), and intermediates like lactate and succinate are represented by large colored arrows. GTBD-Tk inferred taxonomy is included (recently reclassified *Roseburia* species are denoted as Agathobacter). Gene names and abbreviations are also provided in Supplementary Data 6.

Fig. 6 and Supplementary Data 6). In addition, we speculate that MAG196 and MAG191 are perhaps capable of metabolizing elements of the AcGGM fiber such as the α-galactose side-chain or deacetylated manno-oligosaccharides, which was inferred via detected GH36 and GH130 representatives (Fig. 7, Supplementary Fig. 6 and Supplementary Data 6).

Dietary inclusion of AcGGM also unexpectedly led to pronounced metabolic shifts in numerous phenotypically and

phylogenetically diverse microbial populations that are not normally associated with fiber hydrolysis. Mirroring our 16S rRNA gene analysis, MAGs affiliated to the genera *Dialister* (MAG150), *Catenibacterium* (MAG048), and *Megasphaera* (MAG053) were the three highest enriched populations in expression clusters differentially detected in the AcGGM diet (adj. $p < 0.001$), although none were found to encode CE2/CE17-containing mannan PULs (Figs. 3–4, 7 and Supplementary

Data 6). In particular, the MAG048 proteome included a putative sugar phosphotransferase system (PTS) and GH1 phospho-β-glucosidases (EC 3.2.1.86) that are predicted to catalyze the phosphorylation of disaccharides (such as cellobiose and mannobiose[25,26]) and hydrolyze the PTS-transported sugars into D-glucose and D-glucose 6-phosphate. Concomitantly, glycolysis, and acetogenesis pathways from MAG048 were also highly detected in AcGGM-fed pigs (Fig. 7 and Supplementary Data 6). Together, these data suggest that certain *Catenibacterium* populations are advantageously consuming AcGGM-derived disaccharides that were either available as a fraction of the original AcGGM fiber preparation, have been generated via the actions of other fiber-degrading populations, or have become available via new ecological niches that have been created via the AcGGM-derived structural shifts in the microbiome.

Both MAG053 (*Megasphaera*) and MAG150 (*Dialister*) were predicted via our multi-omics approach to metabolize SCFAs such as lactate and succinate that were generated endogenously by *Lactobacillus*- (i.e., MAG013) and *Prevotella*- (i.e., MAG285) affiliated populations, and produce butyrate and propionate, respectively (Fig. 7 and Supplementary Data 6). Neither of these cross-species relationships had any evidence of being directly influenced by AcGGM and are presumably being driven by underlying availabilities of fermentation intermediates (e.g., lactate and succinate) resulting from as-yet uncharacterized niche fluctuations. Collectively, these data highlight that while a highly tailored MDF can directly activate enzymatically equipped populations in an endogenous microbiome, they can also inadvertently alter the metabolisms of other non-mannanolytic populations creating effects that should be quantified to ascertain the true modulation potential of an MDF. The community dynamics that drive fiber-fermentation is not well understood and is likely a contributory factor that restricts broader MDF development.

## Discussion

Here, we have advanced the MDF concept by showing that the defined relationship between fiber and microbe can be used to directly target specific microbiota in highly complex and competitive gut ecosystems. This was achieved by matching the structural features of our characterized AcGGM fiber to a specific genetic locus in our target microbiota, for which there exists comprehensive biochemical information pertaining to the encoded enzymatic functions.

Using integrated multi-omics analysis, we illustrate the impact of the AcGGM structural configuration on microbial uptake and metabolism within the distal regions of the porcine digestive tract, with the key driver of AcGGM selectivity being the presence of acetylations of mannan, as well as carbohydrate composition and size[27]. Preserving the complexity of AcGGM resulted in a highly specific shift in the composition of the colon microbiome from weaned piglets, with increases in the relative butyrate:acetate ratios and no adverse effect on host growth performance or health status. We showed that the AcGGM fiber activated a metabolic response in specific *Roseburia* and *Faecalibacterium* populations in vivo, as it did in the previous in vitro experiments[5,10], with both populations expressing proteins from unique CE2/CE17-containing mannan PULs that are homologous to a biochemically characterized representative in *R. intestinalis* L1-82. In conclusion, our data provide a foundation for modulatory strategies to design and match custom dietary fibers to unique enzymatic features of their target organisms. However, they also underscore the fact that the greater network of interconnected metabolic exchanges and trophic structures inherent to the gut microbiome is highly susceptible to minor dietary interventions. Ultimately,

further research is required to determine whether AcGGM MDFs can confer improvements to animal performance metrics (such as growth and disease resistance) in the context of full-scale livestock production.

## Methods

**Animals, diets, and experimental design**. Animal care protocols and experimental procedures were approved by the Norwegian Animal Research Authority, approval no. 17/9496, FOTS ID 11,314, and treated according to institutional guidelines. A total of 48 crossbred piglets (Landrace × Yorkshire), 24 male and 24 female, with average initial body weight (BW) of 9.8 ± 0.5 kg and weaned at 28 days of age, were sorted by litter, sex, and weight and randomly divided into twelve pens of four animals, with three pens fed each of the four diets. The piglets in each pen were trained to feed at individual stations containing their respective feed. The animals were housed in an environmentally controlled facility with plastic flooring and a mechanical ventilation system. The temperature of the room was maintained at 22 °C.

Piglets were fed cereal-based diets containing increasing levels of AcGGM diets (1, 2, and 4%). Inclusion levels were selected based on in-house experience from previous feeding trials and communication with the feed industry, where an inclusion level of 1–4% of MDF in feed would be realistic. The highest inclusion level at 4% was selected in order to test a dose response. Diets were pelleted with a 3 mm diameter feed formulated to meet the requirements for indispensable amino acids and all other nutrients (NRC, 2012). The composition of diets is listed in Supplementary Table 3. Pigs were fed semi-ad libitum twice a day at a feeding level equal to about 5% of body weight. To evaluate growth performance, the BW of each pig was recorded at the beginning and once a week. Feed consumption was recorded for each individual pig during the experiment to calculate individual weight gain and feed intake. After each meal, feed leftovers were registered, dried, and subtracted from the total feed intake.

**Production of AcGGM**. AcGGM oligosaccharides for the feeding trial were produced from Norway spruce chips milled with a hammer mill to <2 mm size. Wood chips were then steam-exploded on a pilot-scale steam explosion rig (100 L reactor vessel) at the Norwegian University of Life Sciences (NMBU). The steam explosion was conducted in batches of approximately 6 kg dry matter, 14.5 bar pressure (equivalent to 200 °C), with 10 min residence time. The pH in the collected biomass slurry after the steam explosion was ~3.7, which corresponds to a combined severity factor $R'_0 = 1.70$ for the process. The combined severity (as per[28]) was calculated by $R'_0 = (10^{-pH}) \times (t \times e^{(T_{exp}-100)/14.75})$. Steam-exploded wood was collected in 50 L plastic buckets that were topped up with hot (~70 °C) water. The slurry was transferred to a 60 L cider press (Speidel, Germany) and the liquid fraction was pressed out. Milled wood was collected, soaked in hot water again, and pressed for the second time. The liquid fraction was collected and recirculated through a bag filter 50 μm pore WE50P2VWR (Allied filter systems, England) partly filled with the wood particles as a filter aid. Once free of floating wood particles, the liquid fraction of hemicellulose was filtered through a 5-kDa spiral wound Polysulphone/polyethersulphone ultrafiltration membrane, GR99PE polyester (Alfa Laval, Denmark) that was deliberately fouled to prevent larger oligosaccharides from running through the permeate, using a GEA pilot-scale filtration system Model L (GEA, Denmark). The fraction retained by the membrane was concentrated by nanofiltration using a TriSep XN 45, which had a higher efficiency for permeating water. The filtrate was further concentrated by vacuum evaporation (set to 65 °C) and the concentrate was freeze-dried and homogenized with a grain mill. The final product consisted of 0.9% rhamnose, 2.7% arabinose, 13.7% xylose, 58.9% mannose, 14.9% glucose, and 9.4% galactose (determined by gas chromatography as alditol acetates after sulfuric acid hydrolysis[29]). AcGGM contained 0.73 % ash and 2.4% protein (quantified from total nitrogen by the Kjeldahl method). The Man:Glc:Gal ratio in the mannan was 4:1:0.6, and the DA = 0.35[9] (determined by acetate release from NaOH treated AcGGM by the same method as described in the SCFA section below). Moreover, previous two-dimensional (2D) $^1$H-$^{13}$C heteronuclear single quantum correlation (HSQC) NMR analysis of the AcGGM used in this study, showed the distribution of acetylations on mannose units, indicating that spruce mannan hydrolysate without enzyme addition contained prevalently 2-*O*-, some 3-*O*-acetylations, and a lower degree of 4-*O*- and 6-*O*-acetylations[9]. The dry matter content was determined by drying 0.2 g of the sample at 105 °C for 20 h. The remaining sample was burned at 600 °C for 24 h in an oven (Carbolite, Sheffield, England) to determine ash content. All measurements were performed in triplicates.

**Fecal scoring**. During the experiment, fecal consistency was assessed using a scoring system developed by Pedersen and Toft[30] to improve and help standardize current protocols for clinical characterization of fecal consistency. The scoring was based on the following 4 consistency categories: score 1 = firm and shaped, score 2 = soft and shaped, score 3 = loose, and score 4 = watery. Samples with a score 3 or 4 are considered diarrheic. Daily fecal scores for each pen were recorded throughout the trial.

**pH measurements**. The pH of digesta samples from duodenum, jejunum, ileum, cecum, and colon were measured immediately after slaughter. Samples were placed in universal containers and pH measurements made using an Inolab pH7110 pH meter (WTW, Germany).

**Blood sampling and flow cytometry**. Blood samples were collected from the same six piglets per diet at 0, 7, and 27 feeding days. The blood samples were taken 1–2 h post-prandial by venipuncture in the jugular vein while pigs were kept on their backs. Non-heparinized and $K_3EDTA$ vacuum tubes (Beckman Dickson Vacutainer System) were used to recollect serum and whole blood. The serum was isolated immediately by centrifugation at $1500 \times g$ at 4 °C for 15 min. Serum samples were split in PCR-tubes (200 μL) and stored at −80 °C until analysis. Hematological and clinical analyses were performed with an Advia® 2120 Hematology System using Advia 2120 MultiSpecies System Software and clinical chemistry analyses were performed with Advia 1800 Chemistry System (both from Siemens AG Healthcare Sector).

For flow cytometry analysis, whole blood was diluted 1:1 in RPMI 1640 and kept on ice until single-cell isolation. For the isolation of peripheral blood mononuclear cells (PBMCs) blood was purified by centrifugation in a Ficoll gradient (Kreuzer et al. 2012). Then, isolated PBMCs were incubated with Fixable Yellow Dead Cell Stain Kit (Life Technologies, ThermoFisher Scientific Inc.) followed by primary monoclonal antibodies (mAbs), brief incubation with 30% normal pig serum to block Fc-receptors, and finally fluorescence-labeled secondary antibodies (Abcam plc, UK). To detect the intracellular CD3 epitope, surface-labeled cells were permeabilized with Intracellular Fixation and Permeabilization Buffer Set (eBioscience, Affymetrix Inc.) according to the manufacturer's instructions. Labeled cells were analyzed on a Gallios Flow Cytometer (Beckman Coulter, Inc.) and data were processed using Kaluza 1.5 software (both Beckman Coulter, Inc.). Cell gates were designed to select for single and viable mononuclear cells. Defined markers were used to identify the different immune subpopulations. For monocytes, antibodies against CD45, CD3, CD14, CD163, and MHCII were used (dilution listed in 'Reporting summary'). To analyze regulatory T cells (T reg) the following antibodies were used: CD45, CD3, TCR γ/δ, CD4, CD8, FOxp3, and CD25, while CD45, CD8, NKp46, CD4, CD8, Ki67, and CD27 were used to identify T and NK cells (dilution listed in 'Reporting summary'). The gating strategy used for flow cytometric data is presented in Supplementary Fig. 7.

**Analysis of serum cytokines: MULTIPLEX**. Expression of GMCSF, IFNG, IL-1A, IL1B, IL-1RA, IL-2, IL-4, IL-6, IL-8, IL-10, IL-12, IL-18, and TNFα were measured in serum samples using MILLIPLEX MAP Porcine Cytokine and Chemokine Magnetic Bead Panel - Immunology Multiplex Assay (Merck Millipore) following the manufacturer instructions. The measurement was performed using a Bio-Plex MAGPIX Multiplex Reader (Bio-Rad).

**Small intestine morphology**. The samples of the small intestine were collected on days 0 and 28 for the determination of intestinal morphology and integrity. Intestinal morphological measurements included the following indices: villus height (VH), crypt depth (CD), and VH:CD. The mean values of VH, CD, and their ratio were calculated. Histology evaluation was performed by the Veterinary Histophalogy Center, VeHiCe, Chile.

**SCFA analysis**. Samples of digesta from the duodenum, jejunum, ileum, cecum, and colon of individual pigs were collected for SCFA analysis. 250 mg or 250 μL of a sample, depending on the source site were mixed 1:1 with 4 mM $H_2SO_4$, homogenized by shaking at room temperature for 1 h, and centrifuged at $12,000 \times g$ for 10 min. The supernatant was collected with a syringe and filtered through a 0.22 μm pore syringe filter. Samples were stored at −20 °C and centrifuged at $12,000 \times g$ before transferring aliquots into HPLC vials for analysis. SCFA content was analyzed by HPLC using a REZEX ROA-Organic Acid H + (Phenomenex, Torrance, California, USA) $300 \times 7.8$ mm ion exclusion column, isocratic elution with 0.6 mL/min 4 mM $H_2SO_4$ at 65 °C. Eluting analytes were detected by UV at 210 nm. The same HPLC method was used to determine the degree of acetylation, by measuring the acetate released from samples of AcGGM dissolved in 100 mM KOH and relating the values to the mannose content of the sample. All data were analyzed using a two-tailed $t$-test.

**Micro array polymer profiling (MAPP)**. All cell-wall preparations were analyzed similarly to Moller et al.[31]. Briefly, cell-wall glycans were sequentially extracted from 10 mg of each sample by incubation with 2 μg/mL cellulase, pH 8.9 in 20 mM tris buffer for 16 h, 45 °C to release residual glycans bound to cellulose. Each extract was mixed 50/50 with glycerol buffer (55.2% glycerol, 44% water, 0.8% Triton X-100) and spotted with four 5-fold dilutions and two technical replicates onto a nitrocellulose membrane with a pore size of 0.45 μm (Whatman, Maidstone, UK) using an Arrayjet Sprint (Arrayjet, Roslin, UK). The arrays were probed using antibodies from the JIM, LM (Plant Probes), BioSupplies, series and a Carbohydrate Binding Module (CBM, NZYTech) detected using anti-rat (JIM, LM), anti-mouse (BioSupplies) or anti-His (CBM) secondary antibodies conjugated to alkaline phosphatase (Sigma Aldrich)[32]. Microarrays were incubated with 5% milk protein in TBS (tris buffered saline) to block non-probe binding sites, followed by

incubation with primary antibody for 1.5 h. Arrays were washed in TBS, then incubated with secondary antibodies for 1.5 h. After washing with TBS, arrays were developed using an NBT/BCIP substrate to visualize sample spots that contained glycan epitopes that the probes bound to. The NBT/BCIP substrate contained Nitro Blue Tetrazolium (NBT) and 5-bromo-4-chloro-3-indolylphosphate p-toludine salt (BCIP) in Tris buffer (0.5 mM $MgCl_2$, 100 mM diethanol-amine, pH 9.5). The arrays were scanned using a desktop scanner (CanoScan 8800 F, Canon) at 2400 dpi, and the intensity of NBT/BCIP stained spots quantified using Array-Pro Analyzer 6.3 (Media Cybernetics). This produces data for the normalized intensity of spots relative to the background intensity of the nitrocellulose membrane (to control for background staining). We used this data to create a heatmap, for average spot intensity across biological replicates and sample dilutions.

**Microbial sampling**. Fecal samples were collected from six piglets per experimental group ($n = 12$) at days 0, 7, 14, 21, and 27 post-weaning. At the end of the trial, all piglets ($n = 48$) were sacrificed, and distinct samples were collected from the lumen of the duodenum, jejunum, ileum, cecum, and colon. Samples were obtained within the first 15 min after the piglets were sacrificed and the samples were flash-frozen in liquid nitrogen and stored at −80 °C until DNA extraction.

**DNA extraction**. DNA was extracted with a MagAttract PowerMicrobiome DNA/RNA Kit (MO BIO Laboratories Inc., Carlsbad, CA, USA) according to the manufacturer instructions, except for the bead-beating step where we used a FastPrep-96 Homogenizer (MP Biomedicals LLC., Santa Ana, CA, USA) at maximum intensity for a total of 2 min in 4 pulses of 30 s with a 5 min cooling period between each pulse. A KingFisher Flex DNA extraction robot was used for the automated steps of the protocol. The extracted nucleic acids were quantified with a Qubit Fluorimeter and the Qubit dsDNA BR Assay Kit (ThermoFisher Scientific, Waltham, MA, USA) and stored at −80 °C.

**16S amplicon sequencing and analysis**. 16S amplicon sequence data was obtained for all fecal and intestinal samples. The V3-V4 region of the 16S rRNA gene was PCR amplified using the primers Pro341F and Pro805R (Supplementary Table 4), to which the MiSeq adaptors were additionally incorporated on the 5′ ends[33]. The 25 μL PCR reactions consisted of 1× iProof High-Fidelity Master Mix (Bio-Rad, Hercules, CA, USA), 0.25 μM primers, and 5 ng template DNA. PCR thermal cycling began with a hot start step at 98 °C for 180 s and was followed by 25 cycles of 98 °C denaturation for 30 s, 55 °C annealing for 30 s, and 72 °C extension for 30 s, followed by a final, 300 s extension step at 72 °C. Amplicons were individually purified with AMPure XP beads (Beckman Coulter, Indianapolis, IN, USA) and indexed with the Nextera XT Index Kit v2 (Illumina, San Diego, CA, USA) according to the Illumina protocol for 16S rRNA gene sequencing analysis. Next, equal volumes from each indexing reaction were pooled together, and the pool was purified with AMPure XP beads. The purified amplicon pool was then quantified with a Qubit Fluorimeter, diluted, mixed with 15% PhiX Control v3 (Illumina), and denatured according to the aforementioned Illumina protocol. The denatured library was sequenced on the Illumina MiSeq platform using the MiSeq Reagent Kit v3 (600 cycles). Data were output from the sequencer as demultiplexed FASTQ format files.

Processing of the data was done with a combination of standalone programs, QIIME[34] MOTHUR[35], and the R package Phyloseq[36]. To process the data, the paired-end reads for each sample were merged with PEAR[37], specifying a minimum assembly length 400, maximum assembly length 575, minimum overlap 50, and no statistical test. Then, PRINSEQ[38] version 0.20.4 was used to filter low quality reads by requiring a minimum quality score of 10 for all bases and a minimum mean quality of 30. Primer sequences were trimmed in MOTHUR version 1.36.1, and chimeric sequences were identified and filtered out using QIIME version 1.9.1. Next, open reference $OTU_{0.97}$ clustering[39] was performed with VSEARCH[40] version 2.3.2 and the Silva database[41] release 128 as the taxonomy reference. Then, the QIIME core diversity analysis script was run. Differentially abundant phylotypes were identified in both cecum and colon for the control vs. 4% AcGGM samples using both the MetagenomeSeq fitZIG and DESeq2 negative binomial algorithms via the QIIME wrapper. The OTU table, phylogenetic tree, representative sequences, and taxonomy from QIIME were incorporated along with the sample metadata into a Phyloseq version 1.22.3 object in R for data exploration and visualization.

**Whole-metagenome sequencing and analysis**. Whole-metagenome sequencing was performed at the Norwegian Sequencing Centre on 2 lanes of the Illumina HiSeq 4000 to generate $2 \times 150$ paired-end reads. TruSeq PCR-free libraries were prepared for 12 control and 12 AcGGM (4%) samples from the colon. All 24 samples were run in both lanes to eliminate the potential for lane-specific sequencing bias. FASTQ format files were received from the sequencing center, and prior to assembly, these were quality filtered with Trimmomatic[42] version 0.36 whereby TruSeq adaptor sequences were eliminated, sequences were required to have an average quality score above 20, leading and trailing bases with quality below 20 were removed, sequences with an average quality score below 15 in a 4-base sliding window were trimmed, and the minimum read length was required to be 36 bases. Individual sample assembly was accomplished with metaSPAdes[43]

version 3.11.1. MegaHIT[44] version 1.1.3 was used for co-assembly of all 24 samples together as well as co-assembly of the 12 control samples together and the 12 4% AcGGM samples together. MetaBAT[45] version 0.26.3 was used to bin the assemblies, and dRep[46] version 2.0.5 was used to dereplicate the multiple assembly and binning combinations to produce an optimal set of MAGs. MASH[47] version 2.0 used to compare the similarity of the 24 metagenomes by calculating pairwise Jaccard distances which were imported into R for NMDS ordination and visualization. Completeness and contamination were determined for each MAG using CheckM[48] version 1.0.7. Taxonomic classifications of MAGs were performed using GTDB-Tk[49], and relative abundance estimations were generated using CoverM (https://github.com/wwood/CoverM). Feature and functional annotation was completed with the Prokka pipeline[50] version 1.12, and the predicted protein sequences from all 355 MAGs were concatenated to create the metaproteomics reference database. Resulting annotated open reading frames (ORFs) were retrieved, further annotated for CAZymes using the CAZy annotation pipeline with libraries from the July 2018 database release[51,52], and subsequently used as a reference database for the metaproteomics (with the exception of glycosyltransferases).

**Metaproteomics.** Proteins were extracted from each sample by the following method. An aliquot (1 g) of colon digesta from pigs fed either a control diet or a diet supplemented with 4% β-mannan was dissolved 1:1 (w/v) in 50 mM Tris-HCl, pH 8.4. Lysis was performed using a bead-beating approach whereby glass beads (size ≤ 106 μm) were added to the colon digesta slurry and cells were disrupted in 3 × 60 s cycles using a FastPrep24 (MP Biomedicals, Santa Ana, CA, USA). Debris was removed by centrifugation at 16,600 × g for 20 min and proteins were precipitated overnight in 16% ice-cold TCA. The next day, proteins were dissolved in 100 μL 50 mM TrisHCl, pH 8.4, and concentration was determined using the Bradford protein assay (Bradford Laboratories, USA) using bovine serum albumin as a standard. Fifty milligrams of protein were prepared in SDS sample buffer, separated by SDS-PAGE using an Any-kD Mini-PROTEAN gel (Bio-Rad Laboratories, Hercules, CA, USA), and stained using Coomassie Brilliant Blue R250. The gel was cut into 6 slices and reduced, alkylated, and digested[53]. Prior to mass spectrometry, peptides were desalted using C_{18} ZipTips (Merck Millipore, Darmstadt, Germany) according to the manufacturer's instructions.

The peptides were analyzed by nanoLC-MS/MS as described previously, using a Q-Exactive hybrid quadrupole orbitrap mass spectrometer (Thermo Scientific, Bremen, Germany)[54], and the acquired raw data was analyzed using MaxQuant[55] version 1.4.1.2. Proteins were quantified using the MaxLFQ algorithm[56]. Data were searched against a sample-specific database (602.947 protein sequences), generated from the 355 metagenome-assembled genomes (MAGs), and against the genome of a pig (Sus scrofa domesticus) (40.708 sequences). In addition, common contaminants such as human keratins, trypsin, and bovine serum albumin were concatenated to the database as well as reversed sequences of all protein entries for estimation of false discovery rates. Protein N-terminal acetylation, oxidation of methionine, conversion of glutamine to pyroglutamic acid, and deamination of asparagine and glutamine were used as variable modifications, while carbamidomethylation of cysteine residues was used as a fixed modification. Trypsin was used as a digestion enzyme and two missed cleavages were allowed. All identifications were filtered in order to achieve a protein false discovery rate (FDR) of 1% using the target-decoy strategy. A total of 8515 protein groups (20,350 shared proteins) were considered valid, which required the protein group to be both identified and quantified in at least two replicates, and in addition, we required at least one unique peptide per protein and at least two peptides in total for every protein group. In cases where a protein group consisted of two or more homologs protein identifications, both unique and shared proteins are indicated (i.e., Fig. 4c and Supplementary Data 6). The output from MaxQuant was further explored in Perseus version 1.6.0.7 where filtering, data transformation, and imputation were performed.

For metaproteomic enrichment analysis, missing values in the proteomics data were set to zero, and proteins with zero variance across the eight samples, or with a non-zero value in only one replicate animal in either the control or AcGGM treatment were removed. This resulted in 4562 unique protein groups assigned to 12,535 MAG ORFs (i.e., shared detected proteins). A Principle Component Analysis (PCA) was performed on the 4562 × 8 matrix using the R function prcomp. Hierarchal clustering dendrograms were constructed using the R functions hclust with Pearson correlation and method = 'ward.D', and a heatmap was drawn using the function heatmap.2 with scale = 'row'. The number of protein clusters were selected by visual inspection and the clusters were selected using the cutree function. Enzyme Commission (EC) annotation (Supplementary Data 9) and MAG enrichment analysis (Supplementary Data 7) was performed using the hypergeometric distribution function phyper in R with the false discovery rate controlled at 5% using the function p.adjust with method = 'BH'.

**Genome tree.** Phylogenetic analysis was performed using a block of 22 universal ribosomal proteins (30S ribosomal protein L1, L2, L4-L6, L10, L11, L14, L15, L18, and 50S ribosomal protein S3, S5, S7-S13, S15, S17, S19)[57,58]. In addition to the MAGs, we recruited 239 reference genomes for phylogenetic resolution. These genomes were selected based on preliminary examination of the assembled metagenome using metaQUAST[59]. The reference genomes were annotated using the Prokka pipeline in

the same manner as for the MAGs. All identified ribosomal protein sequences were aligned separately with MUSCLE v3.8.31[60] and manually checked for duplications and misaligned sequences. Divergent regions and poorly aligned positions were further eliminated using GBlocks[61], and the refined alignment was concatenated using catfasta2phyml.pl (https://github.com/nylander/catfasta2phyml) with the parameter '-c' to replace missing ribosomal proteins with gaps (-). The maximum likelihood-based phylogeny of the concatenated ribosomal proteins was inferred using RAxML version 8.2.12[62] (raxmlHPC-SSE3 under PROTGAMMA distributed model with WAG substitution matrix) and support values determined using 100 bootstrap replicates. The tree was rooted to the Euryarchaeota phylum and visualized using iTOL[63]. Clades of reference genomes with only distant phylogenetic relation to the MAGs were collapsed to refine the final tree in Fig. 4. The complete tree is available in Newick format as Supplementary Data 4.

**Reporting summary.** Further information on research design is available in the Nature Research Reporting Summary linked to this article.

## Data availability

All sequencing reads have been deposited at the NCBI sequence read archive under BioProject PRJNA574295, with specific numbers listed in Supplementary Table 2. All annotated MAGs are publicly available via https://doi.org/10.6084/m9.figshare.9816581[64]. The proteomics data have been deposited to the ProteomeXchange Consortium (http://proteomecentral.proteomexchange.org) via the PRIDE partner repository[65] with the dataset identifier PXD015757.

## Code availability

The code used to perform the metaproteomic enrichment analysis is available at https://gitlab.com/hvidsten-lab/michalak.

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

## Acknowledgements

We are grateful for support from The Research Council of Norway (Bionær program 244259, 208674/F50, 270038, FRIPRO program, PBP: 250479), as well as the European Research Commission Starting Grant Fellowship (awarded to PBP; 336355 - MicroDE). The sequencing service was provided by the Norwegian Sequencing Centre (www.sequencing.uio.no), a national technology platform hosted by the University of Oslo and supported by the 'Functional Genomics' and 'Infrastructure' programs of the Research Council of Norway and the Southeastern Regional Health Authorities. We thank Sivert Dæhli for performing the detailed analysis of degree of polymerization of the AcGGM.

## Author contributions

L.M., J.C.G., P.B.P. and B.W. conceived the study, performed the primary analysis of the data and wrote the paper (with input from all authors). L.M., J.C.G., L.L., S.L.L.R., L.H.H.,

T.R.H., P.B.P. and B.W. designed the figures. L.L. and M.Ø. designed, performed and analyzed the animal experiments. S.L.L.R., L.H.H., M.Ø.A., T.R.H. and J.D. generated the data and/or contributed to the data analyses. W.G.T.W. and C.T.-J. generated and analyzed MAPP data. N.T., V.L. and B.H. annotated and curated the MAGs and identified carbohydrate-active enzymes.

## Competing interests

The authors declare no competing interests.
