## [Peer Review File · Nature Communications]

REVIEWERS' COMMENTS

Reviewer #2 (Remarks to the Author):

Overall I think the authors have done a very thorough job of responding to the reviewers comments and even carrying out further experiments and I commend them on this.

I think the work provides a uniquely in-depth and state of the art analysis of the effects of fibre supplementation on the endogenous microbiota structure and function and is in this regard of significant interest to many researchers in the microbiota field.

One point I think does need further clarification is regarding the authors interpretation of the effect of supplementation on butyrate levels.

One of the main reasons AcGGM was chosen is stated that it potentially has the ability to specifically promote growth of specific butyrate producers in the gut and this is good as higher butyrate is a good thing for gut health. However the data show the absolute levels of butyrate do not increase significantly (although the trend is increasing butyrate), but the ratio of But:Ac does i.e. all that has happened is acetate levels have dropped.

So statements like Line 158 '...suggesting that AcGGM inclusion promotes butyrogenic fermentation' are not actually true. AcGGM supplementation actually results in lower levels of total SCFAs (driven by lower acetate) vs no supplementation.

In addition the authors have not explored or at least mentioned the effects supplementation has on propionate levels. Why is this? It should at least be shown and ideally discussed.

Minor point:

Fig 7 legend and line 411. Lactate and succinate are not SCFAs. Reword to something like 'SCFAs and intermediates like lactate and succinate'.

Reviewer #3 (Remarks to the Author):

The authors appear to have carefully and respectfully considered each comment from the four reviewers. Their responses are detailed and appropriate, and in many cases include additional work, figures, and/or clarifications. This revised version reads much better and addresses most of the requested changes.

This work is scientifically sound and presents an interesting study of broad interest. The authors have addresses all of my concerns in appropriate detail. There is still some minor issues with the overall novelty and whether the conclusions are a bit over-reaching, but I am satisfied overall with the revisions and do not have further corrections.

Reviewer #4 (Remarks to the Author):

I am satisfied with the changes that the authors have made in response to my comments and suggested edits. I agree that these changes have significantly improved and strengthened the manuscript. I only have a few additional edits suggested below:

Line 164: Legend for Figure 2 needs updating to reflect that this figure now shows both absolute and relative abundances of butyrate

L345: While this figure displays the diversity of organisms and pathways contributing to butyrate production, I would appreciate a clearer delineation between how the substrates (e.g., B-mannans) relate to the GH enzymes and EC numbers. As currently presented, it's unclear which enzymes work on which substrates.

Line 644: given that both 16s rRNA gene sequencing and metagenomic analyses are performed in these papers, I would try to avoid confusion associated with '16s metagenomic analyses' – just refer to it as 16S rRNA gene community analyses or something similar.

Reviewer #5 (Remarks to the Author):

The authors have improved the original manuscript considerably and put in a significant amount of work to address the reviewer's comments.

It is an interesting paper and is very clearly expressed in its current form.

COMMENTS FROM REVIEWER(S):

Reviewer #2 (Remarks to the Author):

Overall I think the authors have done a very thorough job of responding to the reviewers comments and even carrying out further experiments and I commend them on this. I think the work provides a uniquely in-depth and state of the art analysis of the effects of fibre supplementation on the endogenous microbiota structure and function and is in this regard of significant interest to many researchers in the microbiota field.

RESPONSE: We thank reviewer #2 for this very positive comment!

- a) One point I think does need further clarification is regarding the authors interpretation of the effect of supplementation on butyrate levels. One of the main reasons AcGGM was chosen is stated that it potentially has the ability to specifically promote growth of specific butyrate producers in the gut and this is good as higher butyrate is a good thing for gut health. However the data show the absolute levels of butyrate do not increase significantly (although the trend is increasing butyrate), but the ratio of But:Ac does i.e. all that has happened is acetate levels have dropped. So statements like Line 158 ‘...suggesting that AcGGM inclusion promotes butyrogenic fermentation’ are not actually true. AcGGM supplementation actually results in lower levels of total SCFAs (driven by lower acetate) vs no supplementation. In addition the authors have not explored or at least mentioned the effects supplementation has on propionate levels. Why is this? It should at least be shown and ideally discussed.**

RESPONSE: We realize that this was a little imprecisely written and have made some changes to comply with the comments from reviewer #2.

Original text:

“Measurements of major short-chain fatty acids (SCFAs) in the cecum and colon showed a trend of incremental increases of absolute butyrate levels as AcGGM levels were increased (Fig. 2). Butyrate is commonly measured in the colon at levels that are approximately one-third of acetate, the most abundant SCFA. Compared to control animals, the mean molar ratio of butyrate to acetate in samples from pigs fed 4% AcGGM increased in both the cecum (from 0.29:1 to 0.37:1 mM, p-value: 0.03) and colon (from 0.35:1 to 0.45:1 mM, p-value: 0.002) (Supplementary Dataset 1), suggesting that AcGGM inclusion promotes butyrogenic fermentation.”

Changed to (L152):

“Measurements of major short chain fatty acids (SCFAs) in the cecum and colon showed a trend of incremental increases of absolute and relative butyrate levels as AcGGM levels were increased (Fig. 2). However, despite SCFA data suggesting that AcGGM inclusion promotes butyrogenic fermentation, it also showed that there was no statistically significant increase in total SCFA levels (Supplementary Dataset 1). Similarly, the levels of propionic acid were not affected by AcGGM inclusion (Supplementary Dataset 1). While changes in SCFA and microbiome composition (Fig. 2-3) resulted from AcGGM inclusion, we observed no adverse effects on the host’s physiology, with the average weight, feed conversion ratio, blood cell composition, T cell population, and colon morphology not differing between the control and AcGGM treatments (Supplementary Fig. 1, Supplementary Dataset 1).”

Minor point:

b) Fig 7 legend and line 411. Lactate and succinate are not SCFAs. Reword to something like ‘SCFAs and intermediates like lactate and succinate’.

RESPONSE: Thanks for pointing this out.

Original text: “SCFAs (butyrate, acetate, propionate, lactate and succinate)”

Changed to L407: “*SCFAs (butyrate, acetate, propionate), and intermediates like lactate and succinate*”

Reviewer #3 (Remarks to the Author):

The authors appear to have carefully and respectfully considered each comment from the four reviewers. Their responses are detailed and appropriate, and in many cases include additional work, figures, and/or clarifications. This revised version reads much better and addresses most of the requested changes.

This work is scientifically sound and presents an interesting study of broad interest. The authors have addresses all of my concerns in appropriate detail. There is still some minor issues with the overall novelty and whether the conclusions are a bit over-reaching, but I am satisfied overall with the revisions and do not have further corrections.

RESPONSE: We thank reviewer #3 for the positive respond to our efforts of improving the manuscript.

Reviewer #4 (Remarks to the Author):

I am satisfied with the changes that the authors have made in response to my comments and suggested edits. I agree that these changes have significantly improved and strengthened the manuscript. I only have a few additional edits suggested below:

Line 164: Legend for Figure 2 needs updating to reflect that this figure now shows both absolute and relative abundances of butyrate.

RESPONSE:

Original text: “Absolute levels of butyrate detected in the cecum and colon digesta of pigs fed the four different AcGGM diets with varying inclusion levels (0-4%).”

Changed to (L163): “*Butyrate detected in the cecum and colon digesta of pigs fed the four different AcGGM diets with varying inclusion levels (0-4%). a, display absolute levels and b, relative levels of butyrate.*”

L345: While this figure displays the diversity of organisms and pathways contributing to butyrate production, I would appreciate a clearer delineation between how the substrates (e.g., B-mannans) relate to the GH enzymes and EC numbers. As currently presented, it’s unclear which enzymes work on which substrates.

RESPONSE: Thanks for pointing out this unclarity, we have modified the colouring in the figure to group each substrate with their associated CAZymes (e.g. green box for B-mannans) and added an explanation to the end of the legend to **Fig 6:**

L369: “*CAZymes (GH, CE and GT) involved in the catabolism of the listed polysaccharides are grouped according to box colour and their specific activities are described on www.cazy.org. Enzymes*”

associated to acetate and butyrate metabolic pathways are listed as EC (enzyme commission) numbers. Corresponding functions can be found on <https://www.genome.jp/kegg/kegg2.html>.

Line 644: given that both 16s rRNA gene sequencing and metagenomic analyses are performed in these papers, I would try to avoid confusion associated with ‘16s metagenomic analyses’ – just refer to it as 16S rRNA gene community analyses or something similar.

RESPONSE:

Original text: “Illumina protocol for 16S metagenomic sequencing library preparation”

Changed to (L663): “*Illumina protocol for 16S rRNA gene sequencing analysis*”.

Reviewer #5 (Remarks to the Author):

The authors have improved the original manuscript considerably and put in a significant amount of work to address the reviewer's comments.

It is an interesting paper and is very clearly expressed in its current form.

RESPONSE: We thank reviewer #5 for this very positive opinion of the revised manuscript.